# Influence of Maternal and Neonatal Factors on Transplacental Passive Immunity after Vaccination against COVID-19

**DOI:** 10.3390/vaccines12080860

**Published:** 2024-07-31

**Authors:** Rebeca Martínez-Quezada, Omar Esteban Valencia-Ledezma, Tito Ramírez-Lozada, Carlos Emilio Miguel-Rodríguez, Juan Carlos Fernández-Hernández, Gustavo Acosta-Altamirano

**Affiliations:** 1Unidad de Investigación, Hospital Regional de Alta Especialidad de Ixtapaluca, IMSS-Bienestar, Ixtapaluca 56530, Mexico; rebemarq_710@hotmail.com (R.M.-Q.); esteban84valencia@gmail.com (O.E.V.-L.); carlos_emilio_13@hotmail.com (C.E.M.-R.); 2Consejo Mexiquense de Ciencia y Tecnología (COMECYT), Toluca 50120, Mexico; 3Servicio de Ginecología y Obstetricia, Hospital Regional de Alta Especialidad de Ixtapaluca, IMSS-Bienestar, Ixtapaluca 56530, Mexico; titolozada@yahoo.com.mx; 4Dirección de Investigación, Hospital General de México “Eduardo Liceaga”, Ciudad de México 06726, Mexico; fernandezhernadezjc2022@gmail.com

**Keywords:** placental transfer, antibodies against SARS-CoV-2, COVID-19 vaccination, influencing factors, maternal factors, fetal factors, neonatal factors

## Abstract

In the present study, we evaluated the influence of maternal and neonatal factors on the efficiency of the placental transfer of neutralizing antibodies against SARS-CoV-2. Vaccination during pregnancy provides fetal and neonatal protection through the passive transplacental transfer of maternal neutralizing antibodies. To date, little information is available regarding the factors that affect the transfer of antibodies against SARS-CoV-2. A retrospective, cross-sectional, observational, and analytical study was carried out. It was found that several biological factors could be altering transplacental passive immunity after vaccination against COVID-19. In our study population, type 2 diabetes mellitus and chronic hypertension tended to decrease efficiency, while data from women with pre-eclampsia showed better indices compared to those from women with healthy pregnancies. Neonates born prematurely showed lower transfer rates when compared to healthy neonates. The premature rupture of membranes significantly decreased antibody transfer. Taken together, the data suggest that vaccination against COVID-19 during pregnancy is effective even under certain unfavorable clinical conditions for the mother, fetus, and neonate. It is important to create and disseminate immunization strategies in vulnerable populations to reduce maternal and perinatal morbidity and mortality associated with infections preventable by vaccination.

## 1. Introduction

Pregnant women with COVID-19 are a population especially susceptible to developing severe disease, and they may develop complications such as infections, pregnancy-induced hypertension, and admission to intensive care units [1].

As with other maternal infections, the adverse outcomes of SARS-CoV-2 infection may be preventable through vaccination [2]. Thus, the COVID-19 pandemic highlighted the need for immunization in pregnant women [3].

Vaccination during pregnancy can provide one or more of the following benefits: preventing maternal morbidity and mortality, reducing the risk of intrauterine infection and fetal disease, and conferring passive immunity to the neonate [4].

Although early testing excluded pregnant women, immunization against SARS-CoV-2 has been shown to be safe and effective in this specific population and induces a strong binding and neutralizing antibody response, providing protection to both mothers and to newborns during a period of great vulnerability to contracting respiratory infections [5,6].

Vaccination during pregnancy provides fetal and neonatal protection through the passive transplacental transfer of maternal neutralizing antibodies. Before 16 weeks’ gestational age, antibody transfer is minimal, with an increase during the second trimester of pregnancy, reaching a peak during the third trimester, especially during the last 4 weeks of pregnancy, with higher fetal IgG levels than maternal serum levels observed at 40 weeks of gestation, which is why placental transfer ratios are usually greater than 1.0 [1,2].

The effectiveness of passive transplacental transfer has already been demonstrated for several vaccine-preventable viruses, such as hepatitis B, rubella, influenza, and measles [7]. After the COVID-19 pandemic and the beginning of vaccination campaigns against it, several studies evaluated the transplacental transfer of IgG against SARS-CoV-2, finding transfer ratios ranging from 0.34 to 1.9, according to different authors and under varied conditions [7,8,9,10,11,12,13,14,15]. 

In this sense, a previous study, carried out in our laboratory, showed that, in the Mexican population, the neutralizing antibodies generated after vaccination against SARS-CoV-2 are efficiently transferred to the neonate, with a placental transfer ratio equivalent to 1.24 (±0.78) [16]. 

In this study, we did not find statistically significant differences when comparing the number of doses received, the type of vaccine, or gestational age at time of vaccination; however, we observed a considerable variability in the results obtained, suggesting that other factors could be intervening in the transfer efficiency [16]. 

In this regard, it has been shown that optimal placental transfer requires a healthy, full-term pregnancy and a functionally intact placenta; in addition, it has been proposed that several biological factors (maternal and neonatal) can modify the transplacental transfer of immunoglobulins IgG, reducing the effectiveness of vaccines [2,17].

It has been shown that, for various vaccine-preventable viruses, the transfer of antibodies across the placenta can be affected by factors such as gestational age at delivery, birth weight, various maternal infections, some chronic medical conditions, placental anomalies, the total concentration of IgG in maternal blood, the type of vaccine, as well as factors associated with the FcRn receptor, among others [2,17,18]. 

It is important to mention that it is estimated that vaccination during pregnancy continues to be suboptimal. Addressing the concerns of pregnant women and the medical community itself about immunization during pregnancy will allow us to provide accurate data on maternal and neonatal benefits that will allow us to create strategies to minimize the risk of vaccine-preventable infections in mothers and infants.

Therefore, the purpose of the present study is to determine if there are some biological factors (maternal, fetal, and neonatal) that could have a negative influence on transplacental passive immunity after vaccination against COVID-19 in order to, in this way, suggest surveillance and follow-up actions for populations vulnerable to developing severe infections associated with SARS-CoV-2.

## 2. Materials and Methods

### 2.1. Study Design

A retrospective, cross-sectional, observational, and analytical study was carried out. The participants were treated in 2 hospital care centers in the Estado de Mexico, one at the second level and the other at the third level of care, throughout the interval between September 2022 and March 2023, as detailed in a previous study. Acceptable participants included all pregnant women admitted to the Gynecology and Obstetrics Unit for delivery who had a complete or partial COVID-19 vaccination scheme, were disposed to participate, and were able to give informed consent [16]. 

### 2.2. Data Collection

The transplacental transfer ratios of antibodies induced by vaccination against COVID-19 in the study population were previously reported [16]. They were used as a reference for this study; the data were categorized according to maternal age, maternal comorbidities, duration of pregnancy, and type of delivery, as well as complications in pregnancy, in the fetus, and in the neonate. 

Data from 10 patients in the initial cohort were excluded due to a lack of information needed for the study. In this way, 104 women immunized against SARS-CoV-2, without clinical history of COVID-19, who had complete data in the clinical records were included in the study. Of the total pregnancies, 101 were singletons and 3 were twins, allowing 107 neonates to be included in the study.

As previously described, the concentration of antibodies (IgG neutralizing antibodies specific to the RBD domain of the S protein) was obtained by flow cytometry, using the LEGEND-plex™ SARS-CoV-2 Neut Ab Assay (1-plex) w/VbP kit (BioLegend, San Diego, CA, USA, Cat # 741127) according to the manufacturer’s instructions. In brief, an immunoassay is a competitive assay using bead-based technology, precisely designed to measure the interaction between analytes and ACE2 protein. An immunoassay utilizes beads that are conjugated with ACE2 protein. This conjugate serves as the capture component that will bind to the target analyte (e.g., the S1-Fc chimera, which then acts as the detection antibody and is intended to bind to ACE2 in the presence of the analyte). The system is designed such that the presence of an analyte will affect the binding competition between the detection antibody and the neutralization antibodies (anti-human S1 recombinant antibody). As the concentration of the neutralization antibody increases, it will outcompete the detection antibody, leading to a decreased fluorescent signal. Streptavidin–phycoerythrin (SA-PE) is added following the binding reactions. SA-PE binds to the biotinylated S1-Fc detection antibody, leading to a fluorescent signal. The intensity of this signal is inversely proportional to the concentration of the neutralization antibody due to the competitive nature of the assay. The immunoassay generates an inverted standard curve, meaning that lower concentrations of neutralization antibodies produce higher signals, while higher concentrations result in lower signals. This allows for the interpolation of antibody concentrations from the standard curve, using logistic regression, with Legendplex v.2021.07.01 software (BioLegend).

Passive immunity, expressed in the present study as the transplacental transfer ratio, was considered as the ratio of neutralizing antibodies in umbilical cord blood and maternal blood. The expected transfer efficiency was >1.0, indicating higher umbilical cord titers compared to maternal titers at delivery [16].

The clinical and sociodemographic data were obtained through a survey of the participants and/or through the consultation of physical clinical records and the SaludNess platform of the HRAEI (Hospital Regional de Alta Especialidad de Ixtapaluca). All participants gave written informed consent for access to their personal data.

### 2.3. Statistical Analysis

Clinical and sociodemographic data were analyzed using descriptive statistics. The data sets were subjected to the Kolmogorov–Smirnov normality test. Differences between two groups were analyzed using the Mann–Whitney test and differences between three or more groups were analyzed using the Kruskal–Wallis test, followed by Dunn’s post hoc test. Statistical significance was considered for *p* < 0.05. The analysis was carried out using the Prism 8 statistical package (Graphpad Prism, San Diego, CA, USA).

## 3. Results

### 3.1. Clinical and Sociodemographic Characteristics of the Study Population

The mothers included in our study population received between one and four doses of a COVID-19 vaccine. None of the participants had previous SARS-CoV-2 infection. The participants reported being immunized with one of the following vaccines in a homologous or heterologous scheme: AstraZeneca (AZD1222), Pfizer (BNT162b), Gam-COVID-5-Vac (Sputnik V), or Cansino (Ad5-nCoV-S). However, as we previously reported [16], there were no statistically significant differences between the quantity of immunizations, the type of vaccine, or gestational age at the time of immunization, so the present study only considered a group of vaccinated women. 

Data on clinical and demographic characteristics and perinatal outcomes are presented in Table 1.

A total of 104 mothers were included in the study, with a median age of 24.5 years (IQR: 19–32) and a BMI of 28.73 kg/m^2^ (IQR: 24.91–32.2). In total, 62.5% of the participants were healthy women and the rest had one or more comorbidity.

Of the total pregnancies, 101 were singletons and 3 were twins. A total of 80% of the mothers did not present complications during pregnancy, while the remaining mothers presented some type of hypertensive disorder during pregnancy. Most women had abdominal delivery, at term.

Of the 107 neonates, 52.34% were female; the median weight and height were 2950 g (IQR: 2548–3190) and 50 cm (IQR: 47–51), respectively, and the median gestational age was 39 weeks (capurro) (IQR: 37–40).

Fetal complications were present in 26% of cases, while 30% presented one or more neonatal comorbidity.

### 3.2. Influence of Maternal Factors on Transplacental Passive Immunity after Vaccination against COVID-19

Maternal age did not have a significant influence on the transfer of neutralizing antibodies (*p* = 0.5834). Placental transfer ratios were calculated at 1.059 (IQR: 0.7399–1.415), 1.058 (IQR: 0.6818–1.668), and 1.501 (IQR: 0.7541–1.641) for an adolescent, optimal, and advanced age, respectively (Figure 1A). 

Maternal comorbidities did not appear to have a significant effect (*p* = 0.4235) on placental transfer efficiency compared to healthy women (median transfer ratio 1.176); however, mothers with type 2 diabetes mellitus (T2DM) and hypertension had ratios below 1.0 (median 0.8071 y 0.4815, respectively), that is, they showed an inefficient transfer of antibodies against SARS-CoV-2 (Figure 1B).

### 3.3. Effect of Factors Associated with Pregnancy and Childbirth on the Transplacental Passive Immunity after Vaccination Against COVID-19

No statistically significant differences were found in hypertensive diseases of pregnancy when compared with healthy pregnancies (*p* = 0.4507); however, interestingly, women with pre-eclampsia had the highest placental transfer ratio (median 1.476 (IQR: 1.118–1.6642), vs. 1.033 (IQR: 0.6845–1.596) in women without complications) (Figure 2A).

Although full-term pregnancy showed a more effective transfer of antibodies (median 1.142 (IQR: 0.7292–1.618) vs. 0.8294 (IQR: 0.4506–1.476) in preterm pregnancies), neither the duration of pregnancy nor the type of delivery significantly influenced the efficiency of the placental transfer of antibodies (*p* = 0.898 and *p* = 0.2950, respectively) (Figure 2B,C).

### 3.4. Impact of Fetal and Neonatal Factors on the Transplacental Passive Immunity after Vaccination against COVID-19

The premature rupture of membranes significantly decreased (*p* = 0.0223) the efficiency of placental transfer compared to controls without complications (median 0.6792 (IQR: 0.3054–0.7868) vs. 1.183 (IQR: 0.7530–1.643). The other complications observed in the fetuses did not appear to significantly affect the transfer of antibodies (Figure 3A).

As far as the neonatal factors studied are concerned, prematurity showed the lowest transfer ratio when compared to neonates without comorbidities (median 0.8021 vs. 1.158), which may reflect an inefficient placental transfer of antibodies against SARS-CoV-2 (Figure 3B).

## 4. Discussion

Vaccination strategies, arising from the recognition of the maternal and perinatal consequences of infections during pregnancy, are part of routine pregnancy care to prevent infectious diseases in the mother, fetus, and newborn [3,18]. Such strategies are especially important when the response to transplacental passive immunity may not be optimal [17].

The placental transfer of antibodies begins in the first trimester, with approximately 10% of the maternal concentration in the umbilical cord, and increases significantly in the second trimester, when the expression of the neonatal Fc receptor (FcRn) increases in the placenta. The fetal concentration reaches approximately 50% of the maternal concentration at week 30, and it is not until the last 4 weeks of a full-term pregnancy that the fetal concentration exceeds 100% of the maternal concentration, resulting in a transfer index greater than 1.0 [17,19].

It is important to mention that the placental transfer of antibodies has been shown to be affected by the total concentration of IgG in maternal blood. In the present study, we did not evaluate total IgG titers, which would have allowed us to generate a proportion corrected for maternal IgG or a standard cut-off point, allowing us to define defective proportions of maternal–fetal antibody titers in the different conditions.

Nevertheless, some authors have defined cut-off values >1.0 for antigens such as influenza and pertussis in apparently healthy women, as well as transfer indices close to this value for the RBD, Spike, and N proteins in women who developed COVID-19 [20].

On the other hand, in vaccinated women, transfer ratios vary widely depending on the author or the conditions of the study, being generally around 0.4 and 0.7 in women with mRNA vaccines [7,9], and with variations according to the immunization trimester being observed, where, in the third trimester, proportions greater than 1.0 can be reached [10]. 

In this regard, Edlow et al. (2020) suggest that values of approximately 0.7 represent an inefficient transfer of antibodies against SARS-CoV-2 [21]. In this way, we consider, as other authors have proposed, that the cut-off point to consider an efficient transfer is equal to or greater than 1.0 [20,22].

It has been shown that other factors can disturb the transfer of antibodies from the mother to the fetus, potentially reducing the effectiveness of various vaccines [19].

Among the factors that can influence the efficiency of placental antibody transfer, it has been suggested that maternal age, weight, parity, and type of delivery (vaginal or abdominal) could potentially play an important role [17,19]. However, one study showed that none of the aforementioned factors influenced the transfer of rubella IgG [23]. In accordance with the latest results, in our study, we did not find significant differences with respect to maternal age, the presence of overweight or obesity, or the type of delivery (Figure 1A,B and Figure 2B,C).

In contrast, we found that mothers with type 2 diabetes mellitus and chronic hypertension had an inefficient transfer (less than 1.0) of neutralizing antibodies against SARS-CoV-2 (Figure 1B). 

Even though our sample size with the aforementioned comorbidities is very small and we consider it necessary to study a larger population to obtain conclusive results, previous studies with other pathogens suggest that there could be an association with their presence and transfer efficiency. However, the information continues to be scarce and discrepant.

Previous studies have evaluated the impact of such conditions without having conclusive results. For example, some studies demonstrated that, for bacterial pathogens, such as Klebsiella and Pseudomonas, hyperglycemia induced an increase in placental transfer, while another indicated that, in diabetic women, the concentration of total IgG and the transfer of some subclasses of these antibodies is low when compared to non-diabetic women [18,24,25,26]. In this regard, França and collaborators (2012) [24] suggested that placental alterations that facilitate glucose transport in women with diabetes also favor the decrease in the expression of the FcRn receptor in maternal, umbilical cord, and placental blood, explaining the decrease in the subclasses IgG1, IgG3, and IgG4.

Given the prevalence of diabetes and hypertension in Mexico, it is important to conduct studies that determine the association between such comorbidities and the effectiveness of transplacental passive immunity after vaccination against COVID-19.

In this sense, although, to date, there are no studies on transplacental passive immunity, recently, a study carried out in the Mexican population demonstrated that in non-pregnant women and adult men, the concentration of neutralizing antibodies against SARS-CoV-2 did not decrease significantly with the presence of hypertension and diabetes compared to healthy subjects [27]. Subsequent studies should aim to demonstrate whether these antibodies are efficiently transferred across the placenta.

Both diabetes and hypertension can develop during the course of pregnancy, potentially affecting antibody transfer, with a decrease in some IgG subclasses, as mentioned in the previous paragraph. However, our study revealed that women with hypertensive diseases of pregnancy, such as pre-eclampsia, gestational-induced hypertension, and gestational diabetes, efficiently transfer neutralizing antibodies against SARS-CoV-2 to their neonates (Figure 2A).

In this context, it has been suggested that, contrary to what is observed in women with pre-existing diabetes, mild gestational hyperglycemia induces an increase in IgG3 transfer [26]. This could be explained under the premise that hyperglycemia is associated with a variety of placental alterations that possibly facilitate the transport of some immunoglobulins. In addition to this, the greater capillarization of placental villi has been observed in women with mild gestational hyperglycemia, which can facilitate the placental transfer of a variety of substances [18].

One study showed that pregnancy-associated hypertension increases the transfer of IgG against *Klebsiella* [25]. As mentioned by Wilcox and collaborators [18], the above could be considered paradoxical given the immunopathological damage observed in the placenta of hypertensive women.

Although the underlying mechanisms are not known, it is possible that the same causes that lead to the increased placental transfer observed in hypertensive women could explain our findings in women with pre-eclampsia.

Furthermore, preterm birth has been associated with a decreased efficiency of the placental transfer of antibodies against a wide variety of viruses, including those against measles, mumps, chickenpox, rubella, influenza, diphtheria, pertussis, tetanus, polio, and respiratory syncytial viruses [19,28,29,30,31,32]. In agreement, our results demonstrate an inefficient transfer of neutralizing antibodies against SARS-CoV-2 associated with prematurity (Figure 3B).

However, a recent study on vaccination against COVID-19 indicated that no association was found between preterm birth and umbilical cord anti-S antibody levels after adjusting the timing of the vaccine and the number of doses before childbirth [33].

In this sense, it was suggested that maternal antibody concentration is more important than gestational age at delivery in determining neonatal anti-S antibody levels [33].

Previous studies support this statement, although they found lower concentrations of antibodies against measles, mumps, rubella, varicella zoster, whooping cough, diphtheria, tetanus, influenza, and meningitis in premature and very premature neonates [31,32].

Two vaccination strategies have been proposed to protect premature newborns. On the one hand, immunization in early pregnancy guarantees sufficient time for antibody transport to the infant. On the other hand, vaccination in late pregnancy would maximize the peak IgG response with the peak placental transfer to the infant [18].

It is important to know the complete status of the mother and fetus to choose the best strategy that protects the pair and reduces the morbidity and mortality associated with COVID-19 in the most vulnerable populations.

## 5. Limitations of the Study

As mentioned by Kachikis and collaborators (2024) [33], currently, the correlates of protection against SARS-CoV-2 have not been accurately estimated, and it is likely that the transfer properties after vaccination cannot be generalized to other vaccines and other antibodies. More studies are required to understand the maternal and fetal immune response against this virus.

The placental transport of antibodies generated by the different vaccines differs between the types of IgG elicited. Thus, protein vaccines mainly elicit anti-IgG1 and IgG3 antibodies, while polysaccharide vaccines mainly produce IgG2 antibodies. In our study, due to the distribution of the data (the vaccination scheme diversity), we did not adjust the results by type of vaccination, which could be widely relevant [31].

The sample size represented another limitation. It is necessary to conduct a study with a larger population to avoid any possible bias.

## 6. Conclusions

It was found that several biological factors could be altering the efficiency of the placental transfer of neutralizing antibodies against SARS-CoV-2. In our study population, type 2 diabetes mellitus and chronic hypertension showed the lowest transfer efficiency, while data from women with pre-eclampsia showed better ratios compared to those from women with healthy pregnancies, indicating the need to investigate these factors in a larger population to obtain conclusive data on placental transfer efficiency and the possible pathophysiological mechanisms involved. Neonates born prematurely showed lower transfer rates when compared to healthy neonates. The premature rupture of membranes significantly decreased antibody transfer. The data suggest that vaccination against COVID-19 during pregnancy is effective even under certain unfavorable clinical conditions for the mother, fetus, and neonate. It is important to create and disseminate immunization strategies in vulnerable populations to reduce maternal and perinatal morbidity and mortality associated with infections preventable by vaccination.

## Figures and Tables

**Figure 1 vaccines-12-00860-f001:**
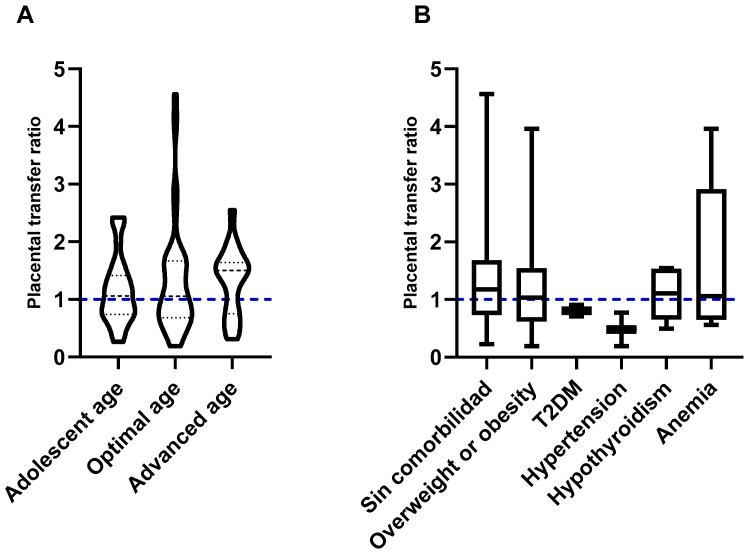
Influence of maternal factors on the transplacental passive immunity after vaccination against COVID-19. (**A**) Influence of maternal age. Maternal age at the time of gestation did not influence the efficiency of the placental transfer of antibodies. (**B**) Influence of maternal comorbidities. Mothers with T2DM and HTN showed the inefficient transfer of antibodies across the placenta. The blue dotted lines represent the expected rate.

**Figure 2 vaccines-12-00860-f002:**
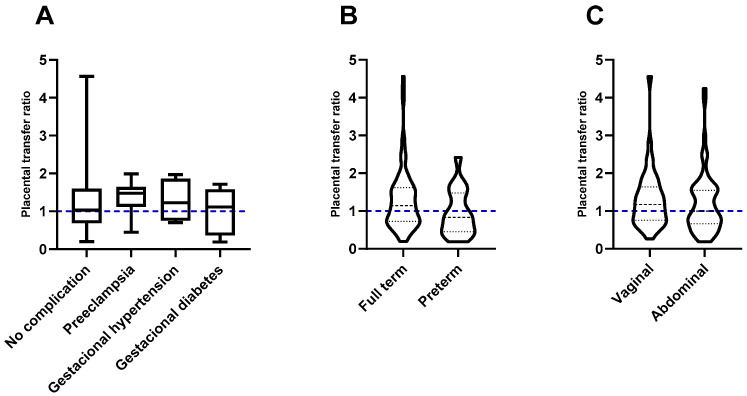
Effect of factors associated with pregnancy and childbirth on the transplacental passive immunity after vaccination against COVID-19. (**A**) Effect of complications in pregnancy. The gestational complications present in the study population had no significant effect on the placental transfer of antibodies; pre-eclampsia showed the highest transfer rates. (**B**) Effect of gestational age. Gestational age did not affect the placental transfer of antibodies. (**C**) Effect of the type of delivery. There were no differences in antibody transfer when vaginal delivery was compared to abdominal delivery. The blue dotted lines represent the expected rate.

**Figure 3 vaccines-12-00860-f003:**
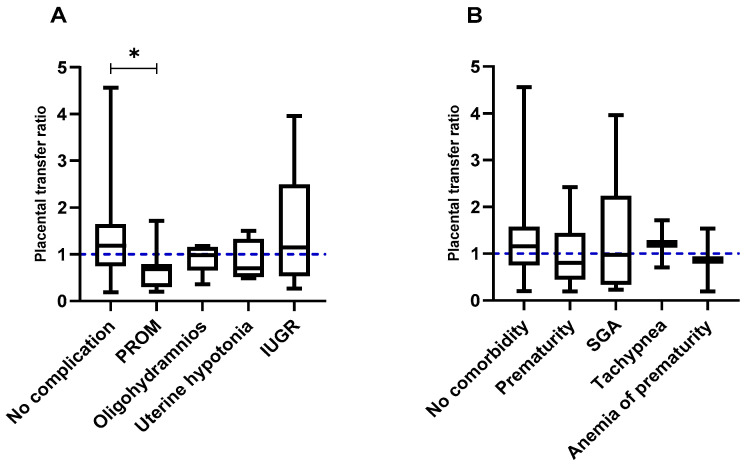
Impact of fetal and neonatal factors on the transplacental passive immunity after vaccination against COVID-19. (**A**) Impact of fetal complications. The premature rupture of membranes significantly impacted placental transfer efficiency (* *p* = 0.0223). (**B**) Impact of neonatal comorbidities. Neonatal comorbidities did not show a significant impact on antibody transfer across the placenta; neonates born prematurely showed the lowest transfer rates. The blue dotted lines represent the expected rate.

**Table 1 vaccines-12-00860-t001:** Clinical and sociodemographic characteristics of the study population.

	N (%)
Maternal Data	
Maternal age	
Adolescent age (≤19 years)	29 (27.88)
Optimal age (20–34 years)	58 (55.77)
Advanced age (≥35 years)	17 (16.34)
BMI	
Normal	67 (64.42)
Overweight	11 (10.58)
Grade I obesity	10 (9.61)
Grade II obesity	10 (9.61)
Morbid obesity	6 (5.77)
Maternal comorbidities	
No comorbidity	65 (62.5)
With comorbidity	39 (37.5)
Overweight or obesity	37 (35.57)
Hypothyroidism	5 (4.8)
Diabetes mellitus type 2 (T2DM)	3 (2.88)
High blood pressure (hypertension)	3 (2.88)
Anemia	5 (4.8)
Pregnancy and Birth Data	
Complications in pregnancy	
No complications	84 (80.2)
With complications	20 (19.2)
Gestational hypertension	4 (3.84)
Pre-eclampsia	8 (7.7)
Gestational diabetes	8 (7.7)
Duration of pregnancy	
Full term	86 (82.7)
Preterm	18 (17.3)
Type of birth	
Abdominal	63 (60.58)
Vaginal	41 (39.42)
Fetal Data	
Fetal complications	
No complications	79 (73.83)
With complications	28 (26.17)
Premature rupture of membranes (PROM)	9 (8.41)
Oligohydramnios	6 (5.6)
Uterine hypotonia	4 (3.73)
Intrauterine growth restriction (IUGR)	9 (8.41)
Neonatal Data	
Gender	
Male	51 (47.66)
Female	56 (52.34)
Neonatal comorbidities	
No comorbidities	74 (69.16)
With comorbidities	33 (30.84)
Prematurity	18 (16.82)
Very premature (28–32 weeks)	3 (2.8)
Moderately premature (32–34 weeks)	5 (4.67)
Late preterm (34–37 weeks)	10 (9.34)
Low weight for gestational age (SGA)	10 (9.34)
Tachypnea	3 (2.8)
Anemia of prematurity	2 (1.87)

## Data Availability

Data are contained within the article.

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
