# Peer review of "Influence of Maternal and Neonatal Factors on Transplacental Passive Immunity after Vaccination against COVID-19"

_vaccines, 2024, doi:10.3390/vaccines12080860_

Round 1
Reviewer 1 Report
Comments and Suggestions for Authors
This report provides the findings of a pilot study concerning the presumed placental transfer of anti-COVID antibodies following previous maternal vaccination. It is clear and easy to follow. I would like to have seen a discussion of the impact of differences in vaccine "life" (6 months, 1 year, etc.) that have haunted anti-COVID vaccines on the timing of maternal vaccination, and the biological effectiveness of the induced fetal antibodies.
Author Response
Comments 1: This report provides the findings of a pilot study concerning the presumed placental transfer of anti-COVID antibodies following previous maternal vaccination. It is clear and easy to follow. I would like to have seen a discussion of the impact of differences in vaccine "life" (6 months, 1 year, etc.) that have haunted anti-COVID vaccines on the timing of maternal vaccination, and the biological effectiveness of the induced fetal antibodies.
Response 1: We appreciate the time dedicated to reviewing this study and your valuable contribution to it. We agree with this comment. In this regard, we would like to be able to complement the manuscript with the information you tell us; However, our study population, for the most part, received a heterologous vaccination scheme, with a large number of possible combinations, with mRNA and viral vector vaccines, making it difficult to perform a conclusive analysis. We are aware of the importance of vaccine life in maternal-fetal immunity, so we will consider this aspect for future research.
The adjustments required by the other reviewers are detailed in the attached manuscript.
Reviewer 2 Report
Comments and Suggestions for Authors
The placental transfer of antibodies is affected by the background IgG immunoglobulin, the time of vaccination and the subclass of antibodies induced by the vaccination. This study presents "real world" results from mothers who had been given a variety of SARS-CoV-2 vaccines in different regimens. It shows that the common conditions encountered in pregnancy do not (as far as maternal/cord ratios show) markedly affect the potential protection of maternal vaccination for the new born in Mexico State. This is despite modestly reduced transfer ratios in subjects with type 2 diabetes and chronic hypertension. There does not appear to be a standard cut-off to define defective the maternal cord ratios or a standard maternal-IgG-corrected ratio but the results show a range similar to that of healthy subjects, including the number of subjects with ranges under 1.0.
The paper of Atyeo C, et al. Compromised SARS-CoV-2-specific placental antibody transfer. Cell. 2021 Feb 4;184(3):628-642.e10. doi: 10.1016/j.cell.2020.12.027) shows a good sample of ratios for different antibodies as well as defective responses so perhaps this could be used as a bench mark.
High maternal IgG immunoglobulin levels are known modulators of the transfer ratio and are though to cause differences in the transfer in different geographical regions so perhaps this should be included in the analysis.
Type 2 diabetes should be written into the abstract no just the abbreviation.
There does not appear to be similar published information from Mexico state in Medline so this will make an original contribution.
Author Response
Comments 1:
The placental transfer of antibodies is affected by the background IgG immunoglobulin, the time of vaccination and the subclass of antibodies induced by the vaccination. This study presents "real world" results from mothers who had been given a variety of SARS-CoV-2 vaccines in different regimens. It shows that the common conditions encountered in pregnancy do not (as far as maternal/cord ratios show) markedly affect the potential protection of maternal vaccination for the new born in Mexico State. This is despite modestly reduced transfer ratios in subjects with type 2 diabetes and chronic hypertension. There does not appear to be a standard cut-off to define defective the maternal cord ratios or a standard maternal-IgG-corrected ratio but the results show a range similar to that of healthy subjects, including the number of subjects with ranges under 1.0.
The paper of Atyeo C, et al. Compromised SARS-CoV-2-specific placental antibody transfer. Cell. 2021 Feb 4;184(3):628-642.e10. doi: 10.1016/j.cell.2020.12.027) shows a good sample of ratios for different antibodies as well as defective responses so perhaps this could be used as a bench mark.
High maternal IgG immunoglobulin levels are known modulators of the transfer ratio and are though to cause differences in the transfer in different geographical regions so perhaps this should be included in the analysis.
Type 2 diabetes should be written into the abstract no just the abbreviation.
There does not appear to be similar published information from Mexico state in Medline so this will make an original contribution.
Response 1:
We appreciate the time dedicated to reviewing this study and your valuable contribution to it. We agree with this comment. Regarding background IgG levels, we can tell you the following: In this study we did not evaluate total IgG titers, which would allow us to generate a proportion corrected by maternal IgG or a standard cut-off point, allowing us to define defective proportions. of maternal-fetal antibody titers in the different conditions.
However, some authors have defined cut-off values ​​>1.0 for antigens such as influenza and pertussis in apparently healthy women, as well as transfer indices close to this value for the RBD, Spike and N proteins in women who developed COVID-19 disease. , (doi: 10.1016/j.cell.2020.12.027).
On the other hand, transfer rates vary widely depending on the author or the conditions of the study, being around 0.4 and 0.7 in women with mRNA vaccines (DOI: 10.1093/cid/ciab266, 10.1016/j.ajogmf.2021.100492), and observing variations according to the vaccination quarter (doi: 10.1016/j.cmi.2021.10.003), where in the third quarter proportions greater than 1.0 can be reached (doi: 10.1016/j.cmi.2021.10.003). In this regard, it is worth mentioning that Edlow et al., (2020) suggested that values ​​of approximately 0.7 represent an inefficient transfer of antibodies against SARS-CoV-2 (doi: 10.1001/jamanetworkopen.2020.30455). In this way, we consider, as had been suggested, that the cut-off point to consider an efficient transfer is equal to or greater than 1.0 (doi: 10.1016/j.cell.2020.12.027, doi: 10.1038/s41467-022-31169 -8).
The above was added in Section 4 (line 233).
In another order of ideas, the term “diabetes mellitus type 2” was defined in the summary, avoiding the abbreviation (line 21).
Reviewer 3 Report
Comments and Suggestions for Authors
The data obtained by the authors show that vaccination against COVID-19 during pregnancy is effective even under some unfavorable clinical conditions for mother and fetus, including maternal hypertension, fetal prematurity, premature placental abruption, which in this case only slightly worsen the transplacental transport of neutralizing IgG. A less expected result of the study was that data from women with pre-eclampsia showed better rates of transplacental IgG transport specific for SARS-CoV-2 compared to women with a healthy pregnancy. Thus, the authors' results are relevant and were obtained using standard methods of clinical, laboratory, and statistical analysis. Meanwhile, I have two minor comments:
(1) Introduction. The purpose of the study should be stated more clearly and specifically.
(2) Section 2.2. It is desirable to clarify that IgG neutralizing antibodies specific to the RBD domain of the S protein have been identified.
Author Response
Comments 1:
The data obtained by the authors show that vaccination against COVID-19 during pregnancy is effective even under some unfavorable clinical conditions for mother and fetus, including maternal hypertension, fetal prematurity, premature placental abruption, which in this case only slightly worsen the transplacental transport of neutralizing IgG. A less expected result of the study was that data from women with pre-eclampsia showed better rates of transplacental IgG transport specific for SARS-CoV-2 compared to women with a healthy pregnancy. Thus, the authors' results are relevant and were obtained using standard methods of clinical, laboratory, and statistical analysis. Meanwhile, I have two minor comments:
(1) Introduction. The purpose of the study should be stated more clearly and specifically.
(2) Section 2.2. It is desirable to clarify that IgG neutralizing antibodies specific to the RBD domain of the S protein have been identified.
Response 1:
We appreciate the time dedicated to reviewing this study and your valuable contribution to it. We agree with your comments.
With respect to the purpose of the study, this study aims to determine if some biological factors (maternal, fetal and neonatal) can have a negative influence on the efficiency of placental transfer of neutralizing antibodies against SARS-CoV-2 to, in this way, Likewise, suggest surveillance and monitoring actions for populations vulnerable to developing severe infections associated with COVID-19. This paragraph was included in Section 1 (line 82).
Updated Section 2.2, noting that the antibodies used were anti-S specific antibodies (line 107). Likewise, the methodology used is described more broadly for greater understanding.